# Effect of Smartphone App’s Intervention on Consumers’ Knowledge, Attitude, Practice, and Perception of Food Poisoning Prevention When Dining Out at Selected Rural Areas in Terengganu

**DOI:** 10.3390/ijerph181910294

**Published:** 2021-09-29

**Authors:** Nur Afifah Mursyida Zaujan, Asma’ Ali, Malina Osman, Hui Yee Chee, Nur Raihana Ithnin, Norashiqin Misni, Surianti Sukeri, Christie Pei-Yee Chin

**Affiliations:** 1Faculty of Fisheries and Food Science, Universiti Malaysia Terengganu, Kuala Nerus 21030, Terengganu, Malaysia; nurafifah.mursyida@gmail.com (N.A.M.Z.); asma.ali@umt.edu.my (A.A.); 2Department of Medical Microbiology and Parasitology, Faculty of Medicine and Health Sciences, Universiti Putra Malaysia, Serdang 43400, Selangor Darul Ehsan, Malaysia; cheehy@upm.edu.my (H.Y.C.); raihana@upm.edu.my (N.R.I.); norashiqin@upm.edu.my (N.M.); 3Department of Community Medicine, School of Medical Sciences, Universiti Sains Malaysia, Kota Bharu 16150, Kelantan, Malaysia; surianti@usm.my; 4Faculty of Computing and Informatics, Universiti Malaysia Sabah, Kota Kinabalu 88300, Sabah, Malaysia; peiyee@ums.edu.my

**Keywords:** food poisoning prevention, mobile application, knowledge, attitude, practice, perception, rural area

## Abstract

(1) Background: Lack of food safety awareness and preventive behaviour when dining out increases the risk of food poisoning. Furthermore, food poisoning cases among rural communities have been rising in recent years. However, the health-related mobile application is a promising tool in improving food poisoning prevention knowledge, attitude, practice, and perception (KAP2) among consumers. Therefore, the current study developed a novel smartphone app, MyWarung©, and determined its efficacy in increasing awareness, attitude, practice, and perception of food poisoning and its prevention when dining out, especially among rural consumers. (2) Methods: A quasi-experimental pre-and post-intervention study with a control and intervention group were performed on 100 consumers in Terengganu. (3) Results: The intervention’s inter-group outcomes were analysed using the Mann–Whitney test, while the within-group effects were ascertained using the Wilcoxon sign rank test via the SPSS software. It was found that the control group had higher median scores in knowledge (30.0, IQR 7.0), attitude (46.0, IQR 5.0), and practice (34.0, IQR 3.0) than the intervention group before intervention. After the intervention programme, the intervention group showed significant improvement in food poisoning knowledge (*p* = 0.000), attitude (*p* = 0.001), and practice (*p* = 0.000). However, the intervention group’s perceived barriers (*p* = 0.129) and susceptibility (*p* = 0.069) and the control group’s perceived barriers (*p* = 0.422) did not show any significant improvement. (4) Conclusion: The findings indicated that the MyWarung© mobile app usage enhanced the food poisoning knowledge, preventive attitude, and practice among consumers when dining out.

## 1. Introduction

Food poisoning refers to diseases from food consumption due to the toxins produced by infectious organisms [1]. Common food poisoning bacteria include *Salmonella* spp., *Escherichia coli*, *Campylobacter*, *Staphylococcus aureus*, and *Clostridium botulinum* [2]. Food poisoning symptoms can range from mild to severe, including fever, diarrhoea, vomiting, nausea, and stomach aches [3]. The prevalence of food poisoning as a health problem is growing at an alarming rate worldwide. An estimated 600 million people become ill every year, and 230 thousand people have died due to food contamination [4]. A lack of awareness regarding the severity of this disease continues to increase food poisoning cases in Malaysia [5], with the highest incidence rate of 45.71 in 100,000 Malaysians in 2018 [6]. The increase in food poisoning cases reported over the years reflects Malaysia’s poor food safety situation [7]. Besides, food poisoning cases among rural communities have become more severe due to a lack of public health awareness about food poisoning and its prevention [8,9], indicating the need for an educational programme targeting consumers, especially in rural areas.

Various contributing factors of food poisoning have been identified, the most important being inadequate food safety practices [1]. Most food poisoning cases happen because of poor hygiene and often involve restaurants and market stalls [10]. Nevertheless, dining out is part of the local lifestyle, which has led to the diversity of Malaysia’s foodservice industry [11]. Furthermore, Malaysians tend to prioritise food taste rather than food safety [7]. Greig et al. [12] reported that 97% of food poisoning cases were due to improper food treatment, poor hygiene, and dirty utensils and equipment in the food preparation process. These factors are directly related to food handlers and cause food poisoning among consumers. Ruby et al. [13] stated that consumers are highly susceptible to food poisoning since they are the end-users in the food processing chain.

Dining out is the major contributor to the rise in Malaysian food poisoning cases among consumers [10]. This activity involves eating food outside or buying meals from outside and consuming them at home [14,15]. Many Malaysians dine out daily, encompassing 86.8% of the population, compared to 13.2% who cook and dine at home [16]. Besides, previous studies have reported that lack of food safety awareness has become a significant contributor to the increasing number of food poisoning cases among consumers in Malaysia [17,18]. To mitigate this problem, consumers must enhance their knowledge, attitude, practice, and perception (KAP2) on food poisoning and its prevention when dining out. Therefore, it is necessary to educate them about food premise selection and proper food safety and hygiene guidelines to prevent food poisoning when dining out.

In the past decade, there has been an increasing trend in KAP2 studies. Most of them included food handlers, while few were conducted among consumers [13,18], especially when dining out. Contrary to consumers, food handlers have presumably better food safety practices because of their training and are supervised by the authorities [13,19]. Furthermore, customers are more susceptible to food poisoning due to their lack of knowledge and awareness about food hygiene compared to food handlers. Ali et al. [20] reported that a lack of knowledge and preventive behaviour concerning food safety and hygiene might result in food poisoning and death among consumers because they are at the end of the food supply chain interaction. Therefore, food safety knowledge and practices play essential roles in foodborne diseases prevention [21] as they can influence consumers’ willingness to adapt and adopt the recommended healthy practices [22].

Food poisoning education is necessary for consumers, especially in rural areas. A cross-sectional study by Abdullah and Ismail [23] on food poisoning outbreaks in rural areas of Terengganu demonstrated that consumers are at a higher risk of food poisoning outbreaks; thus, food poisoning prevention in rural areas, particularly in Terengganu, is vital. Besides, a lack of food poisoning knowledge and preventive attitude and practices among rural communities was reported by Bisholo et al. [8], where food is commonly sold at informal outlets and street vendors. This situation is a public health concern since only 17.2% of rural consumers worried about the safety of food prepared away from home, while 52.9% were not bothered. Similarly, Adebowale and Kassim [24] reported on poor food safety practices among rural consumers. These studies showed that food poisoning among rural communities has become more severe due to the lack of public awareness and prevention, indicating the need for an education programme for consumers when dining out, especially in rural areas.

Education is one of the primary interventions to increase consumers’ knowledge and awareness about food poisoning [20]. Nevertheless, a study by Soon [25] found that there have been reports of fake food safety news being shared via social media, causing panic among consumers, and 62% agreed that social media would help avoid food poisoning cases. These findings implied that reliable food safety information is needed to raise consumer awareness and improve food safety practices. Therefore, the current study developed a mobile application as an easy-access information tool containing guidelines for preventing food poisoning when eating out. Besides, the current study aimed to educate consumers when dining out due to the absence of a systematic consumer education programme on food safety and hygiene in East Malaysia [26].

Recent research has improved the use of the smartphone application as an educational tool, as people spend more time on their mobile devices [27]. Moreover, current technologies such as smartphones and activity trackers offer realistic and dynamically tailored approaches that integrate successful behaviour-changing strategies [28]. As a result, mobile health apps have become increasingly popular, with up to 500 million users globally [29]. According to Proudfoot et al. [30], people are interested in using these apps because of their convenience and easy management, and because they can improve their well-being and self-awareness, suggesting the usefulness and practicality of smartphone apps in improving human health and disease prevention. Besides, Middelweerd et al. [28] reported that systematic reviews found app-based approaches to benefit health changes. They empower smartphone users to increase their knowledge and offer solutions through the development of telecommunications and related technologies [31].

MyWarung© is an innovative solution to prevent food poisoning when dining out in Malaysia, particularly in Terengganu. More often than not, consumers do not practice food poisoning prevention when dining out—for example, selecting clean food premises and food handlers. The MyWarung© app educates consumers to observe the food premises’ cleanliness and food handlers’ hygiene before dining. Moreover, consumers can easily access the in-app food poisoning prevention guidelines at their convenience. This research is the first mobile-app-based study that covers the understanding of food poisoning and preventive behaviour when eating out and reports directly to the Ministry of Health of Malaysia (MOH). The app’s objectives are to increase the knowledge and self-awareness of food-poisoning-preventive behaviour among consumers. Consumers who actively engage with the application are likely to experience preventive behavioural changes, as reported by McKenzie et al. [32].

The evidence from several previous studies showed positive results in using smartphone apps as an educational tool [32,33,34]. The current study developed the MyWarung© smartphone app to increase the KAP2 towards food poisoning and its prevention among consumers when dining out. Meanwhile, the focus of this study is to use a smartphone app to deliver self-management education regarding food poisoning prevention among consumers when dining out, especially in rural areas.

## 2. Research Methodology

### 2.1. Study Location

Kuala Nerus, Terengganu was selected as the research site because the Ministry of Domestic Trade and Consumer Affairs reported a rise in food poisoning cases in 2018 [35]. The main city of Kuala Nerus is Kuala Nerus town, which includes other townships like Gong Badak, Seberang Takir, Batu Rakit, Batu Enam, and Tepoh [36]. The Kuala Nerus Administrative Centre houses district-level government offices, financial institutions, business facilities, and recreational areas that are strategically located and accessible [37]. Today, Kuala Nerus consists of many small villages surrounded by agricultural areas, forests, and water bodies [38]. The Malaysia Ministry of Rural Development has defined rural regions as low-density areas including villages, orchards, small settlements, agricultural areas, forests, and water bodies such as rivers, beaches, and lakes, with a population density of less than 10,000 [39]. Moreover, most local village residents are involved in agriculture and fisheries [40]. Besides, rural areas have also been described as less developed villages or small towns [41,42].

With regards to the usage of smartphones among rural communities, a survey done by the Malaysian Communication and Multimedia Commission reported that smartphone users in Malaysia primarily consisted of students (95.5%), pensioners (44.7%), unemployed (56.6%), employed (82.7%), self-employed (78.2%), those with a relatively low income (81.8%), and those in rural areas (67.3%) [43]. Besides, the smartphone penetration in Terengganu was the highest compared to the other states in 2017, with a 133.1 penetration rate per 100 inhabitants [43]. Although Kuala Nerus is considered a rural area, the number of food premises has increased tremendously because of the lifestyle changes within the community. Therefore, the villages in Kuala Nerus were chosen as sampling sites for the present study.

### 2.2. Study Design

The study design and protocol were reported according to the standard protocol items of Transparent Reporting of Evaluations with Nonrandomized Designs (TREND) and Consolidated Standards of Reporting Trials of Mobile Health Application and Online Telehealth (CONSORT E-HEALTH) guidelines [44,45,46].

This research was conducted in Kuala Nerus, Terengganu, from March to June 2020. A quasi-experimental pre-and post-intervention study design were utilised, involving an intervention and control group, because this was a population-based study; thus, it was not feasible to randomise the samples at the individual level [34]. Similarly, Alzoubi et al. [47] opted out of randomising the intervention due to locations and subjects that were difficult to randomise. Therefore, the quasi-experimental design was categorised as a non-randomised trial study design, which compares the intervention group with the control group (no intervention) and is sometimes called pre-post intervention. This is often used to evaluate the benefits of a particular intervention [48].

### 2.3. Study Sampling

The study’s sampling began with the selection of villages, the allocation of villages into intervention and control groups, and the selection of respondents. The researcher selected the most similar socio-economic and population characteristics among the participating villages to ensure a fair comparison between groups. According to a previous study, rural communities are linked to low wages, unemployment, and little or no formal education [24]. Furthermore, the rural population has a significantly higher poverty rate than the urban population, with Malaysia’s rural population declining from 31.6% in 2007 to 23.4% in 2019, since more people migrate to urban areas in search of better economic opportunities [49]. These socio-economic and population characteristics resulted in the inclusion of six villages with fewer than 10,000 residents. Most residents in these villages were self-employed and unemployed, with low monthly incomes and educational levels. Furthermore, their residential areas were separated and surrounded by forest, and most residents were involved in agriculture and fisheries.

A simple random sampling method was used to allocate six villages into the intervention and control groups, using the online number generator OpenEpi: Open Source Epidemiologic Statistics for Public Health (accessed on 20 January 2020 at https://www.openepi.com/Random/Random.htm) to assign random numbers for the villages. Three villages were labelled as the intervention group: Kampung Pengkalan Arang (*n* = 9), Kampung Bukit Tunggal (*n* = 21), and Kampung Tuan Mandak (*n* = 20); while three other villages were included in the control group: Kampung Tebauk (*n* = 17), Kampung Pak Katak (*n* = 20), and Kampung Batu Enam (*n* = 13). Intervention and control group respondents were chosen from separate villages to prevent cross-contamination or interactions between both groups to ensure the intervention programme’s effectiveness [50].

The respondents were selected from each village using convenience sampling to reach 100 participants, with 50 in each intervention and control group village, based on the inclusion and exclusion criteria. However, the respondents in each village were not homogeneous due to their voluntary nature and respondents’ availability. Figure 1 shows the map view of six selected villages in the current study.

The sample size was calculated using the formula suggested by Attri and Kaur [51] and Charan and Biswas [52]. Taking a standard deviation (SD) of handwashing practice of 24.17 in the intervention group, the difference between the mean (d2) of the practice score between the intervention and control group of 14.30, the power of 0.8, type 1 error of 0.05, and the ratio between the intervention and control group of 1:1, the minimum sample size needed in this study was 45 respondents per group [53]. After 10% attrition was considered in the calculation, the final minimum sample size was 50 respondents per group, making 100 respondents in this study.

The inclusion criteria for the participants included purchasing outside food at least once a week, age 18 and above, readiness for 12 weeks of study, willingness to use a customised mobile app if selected in the intervention study, and the ability to read and write in the Malay language since the questionnaire was provided in the Malay language. Meanwhile, the exclusion criteria for this study were refusal to participate and inability to consent from participants.

### 2.4. Instruments

#### 2.4.1. MyWarung© Mobile Application

MyWarung© development began with an idea description, storyboard development, approval of contents by the research team, finalisation of contents in Android Package File (APK) format, publication in the Google PlayStore, and copyright acquisition from the Intellectual Property Corporation of Malaysia (MYIPO). Figure 2 depicts the overall development process for the MyWarung© mobile application. First, the workflow for producing a mobile app started with the idea description for preventing food poisoning. According to a recent study by Mu et al. [54], an ideal mobile app should be simple, flexible, and functional, and provide reliable information to the consumers. Furthermore, science-based and practice-based knowledge is necessary to provide appropriate guidance to consumers in changing their eating behaviours [54].

MyWarung© provides science-based information on food poisoning, including causative agents, symptoms, complications, causes, and prevention. However, science-based awareness alone is insufficient to change consumers’ ingrained daily habits [55]. Thus, MyWarung© offers practice-based information such as a list of BeSS (clean, safe, and healthy) food premises, food premise assessment as a guideline in selecting clean food handlers and food premise, and a platform to lodge a report directly to the Malaysian Ministry of Health when contracting food poisoning after dining out or coming across dirty food premises.

The content of the app was outlined and continuously reviewed by the research team. The six elements and features of food poisoning: introduction, causes, prevention, BeSS (clean, safe, and healthy) food premises, food premise assessment, and complaint to Malaysia’s Ministry of Health (MOH) were recommended by the researcher to be included in the MyWarung© and approved by the research team. On top of that, another essential feature, food poisoning treatment, was included in the app later on. It was suggested and clarified by one of the research team members, a medical doctor with experience in epidemiology, because most customers are unaware of the appropriate treatment for food poisoning. Therefore, the MyWarung© app currently offers a total of seven main features.

The information used in developing the mobile application was retrieved from trusted sources. First, the information for the introduction, causes, preventive measures, and treatment of food poisoning modules were obtained from MyHealth Portal: Ministry of Health, Malaysia [56] and reviewed by a medical doctor with expertise in epidemiology. Next, the list of BeSS food premises and the Public Complaints Management System (SISPAA) provided by the Malaysian Ministry of Health was introduced by the Chief Assistant Director of the Food Safety and Quality Division (BKKM) from the Terengganu State Health Department. Then, the food premise assessment module information was obtained from MyHealth Portal: Ministry of Health, Malaysia [57]. These references assisted the researcher in drafting the storyboard and contents for each of the modules in the MyWarung© mobile app.

The modules’ contents were drafted using Microsoft PowerPoint and presented to the research team to finalise before being incorporated into the MyWarung© app. In addition, the technical problems were also discussed and addressed through meetings and discussions. Furthermore, the images used in the MyWarung© app were inspected thoroughly to ensure that they accurately reflected the content and were comprehensive for Malaysian users. Lastly, after five corrections, the research team approved the final draft MyWarung©’s poster and video contents for educational purposes, focusing on the content’s readability and simplicity.

After that, the researcher discussed the finalised MyWarung© storyboard contents with the mobile application developer, Skyrem Brilliant Service Malaysia. The developers were also enlightened about the functionality and sequence to illustrate how users would explore the app, including the app interface. The developer took three months to finish developing the mobile application. Then, they provided the researcher with the Android Package File (APK) to check for any corrections and additional information to be included in the app. Next, the research team conducted a detailed review of the MyWarung© app until it was finally ready to be published in the Google PlayStore (accessed on 25 June 2021 at https://play.google.com/store/apps/details?id=com.skyrembrilliant.mywarung). Lastly, the copyright for MyWarung© was obtained from the Malaysian Intellectual Property Corporation (MYIPO) with notification number CRLY00025199. Figure 3 illustrates an overview of the MyWarung© landing page.

The MyWarung© app was developed for the Android platform since it is the most popular mobile device operating system among Malaysians [5,33] and cost-effective [58]. The medium of this app is *Bahasa Melayu* to make it understandable for users of various education levels, particularly consumers residing in rural areas of Terengganu.

The first module helps consumers understand the causative agents and complications in food poisoning. Next, the causes of food poisoning and food signs segment discusses the average duration and onset of symptoms caused by foodborne germs and common food sources that cause food poisoning. Measures to prevent food poisoning, food handlers’ ethics, steps to identify spoiled food, and behaviours at food premises are provided in this section. Each sub-module in the food poisoning prevention training is presented in the form of posters and videos. Moreover, the users can find BeSS food premises by typing the district name in the “Search” box. On top of that, each BeSS food premise listed in the MyWarung© app is directly linked to Google maps.

The food premise assessment was designed to help consumers decide whether their eatery of choice is clean or dirty, based on the following star ratings: 1 star = very dirty, 2 stars = dirty, 3 stars = moderately dirty, 4 stars = clean, and five stars = very clean. Last but not least, consumers can directly lodge a report with the authorities by clicking on the complaint to MOH section linked to the Public Complaints Management System (SISPAA) when they contract food poisoning after dining out or discover unhygienic food premises.

#### 2.4.2. KAP2 Questionnaire

The KAP2 self-administered questionnaire was adapted from previous studies and developed with modifications for this research. A total of 42 questions about food poisoning and its prevention were evaluated across six main domains: disease aetiology (4 items), high-risk foods (10 items), food poisoning signs and symptoms (10 items), food poisoning complications (5 items), food spoilage detection (3 items), and food poisoning prevention (10 items) [10,13,59]. The options for knowledge evaluation were “yes,” “no,” and “not sure.” A score of “1” was awarded to the respondent for every correct answer, while a “0” was given for every incorrect, unsure, and unanswered question. Thus, the lowest score is 0, and the maximum score is 35.

The food poisoning prevention attitude included ten questions based on Ismail et al. [60] and Nik Rosmawati et al. [61] to assess respondents’ cognitive behaviour. Participants were asked to rate their level of agreement or disagreement on a 5-point scale: 1 = strongly disagree, 2 = disagree, 3 = neither agree nor disagree, 4 = agree, and 5 = strongly agree. The minimum score is 10, while the maximum score is 50. The construct of food poisoning prevention practices when dining out consisted of 10 items considered key prevention and risk reduction practices for food poisoning [21]. The 4-point Likert scale is scored from “1” to “4”: 1 = never, 2 = rarely, 3 = occasionally, and 4 = always. All points were summed up, and negative items were reverse-scored. The minimum score was 10, while the maximum score was 40.

The perception of food poisoning prevention consisted of five questions that assessed respondents’ barriers (3 items) and susceptibility (2 items) [62]. A 5-point Likert scale was used: 1 = strongly disagree, 2 = disagree, 3 = neither agree nor disagree, 4 = agree, and 5 = strongly agree. The minimum score of perceived barriers is 3, while the maximum score is 15, whereas the minimum score of perceived susceptibility is 2, while the maximum score is 10. The reliability analysis was conducted on the construct of attitude, practice, and perception using Cronbach’s alpha. Based on the findings, high reliability was recorded for attitude (0.848), while practice (0.780) and perception (0.611) had acceptable reliability.

### 2.5. Recruitment and Data Collection

The study was conducted in Kuala Nerus, Terengganu, where pre-and post-intervention measurements were collected from March to June 2020. The study protocol consisted of recruitment and screening based on the inclusion and exclusion criteria, baseline assessment as a pre-test, intervention stage (12 weeks), and follow up as a post-test stage (Figure 4).

In the first stage of participants’ recruitment, the researcher scheduled a meeting with the respective Village Community Management Council (MPKK), where the head of villages collaborated with the research team to recruit participants and brief them about the study objectives. The consumers were provided with a flyer that described the study details and respondents’ eligibility criteria. Then, the selected consumers were invited to assemble in their respective council halls for registration and assigned numbers after their contact details (i.e., name, phone numbers, locality) were recorded. All participants included in the study were informed of the study aims and goals through the study’s subject information sheet and required to provide a written consent upon participation.

Then, the participants (intervention and control groups) were asked to complete a validated pre-intervention KAP2 self-administered questionnaire. After that, the intervention was given to the intervention group. The participants in both groups were blinded to their study condition, so they did not know whether they were in the intervention or control group. According to Karanicolas et al. [63], blinding participants is important because participants who are aware that they did not receive any intervention are more likely to obtain additional knowledge from other sources, which will impact the study’s calculated outcome.

After 12 weeks of intervention, a text message containing a link to the post-intervention KAP2 Google form was sent to both groups to be completed [28]. The control group was later introduced to the MyWarung© app for ethical reasons after all the respondents had submitted the post-intervention KAP2 questionnaire [34,64]. Each respondent was gifted with a token of appreciation upon completing the questionnaire.

### 2.6. Intervention Protocol

The intervention focused on the KAP2 of food poisoning and its prevention whendining out, with the aims of increasing knowledge among rural consumers about food poisoning and its prevention, exposing them to preventive behaviour when dining out, and increasing awareness regarding the cleanliness of food premises and food handlers’ hygiene. Participants can easily access information concerning food poisoning and its prevention by simply referring to the posters and videos in the mobile app. The intervention was implemented for 12 weeks (April to June 2020) using the seven main modules included in the finalised version of the MyWarung© app.

First, the mobile app’s link was provided to the respondents via text message to access the application. Then, the researchers performed and tracked the tutorial about installing and using the MyWarung© app. Once installed, participants in the intervention group were briefed about the app’s features, objectives, and intervention period. Secondly, the participants independently implemented the information provided in the MyWarung© app module in their daily life to improve their preventive behaviour when eating out. Moreover, the researchers created a WhatsApp’s chat group to communicate with participants within the 12-week intervention period [65,66]. They were instructed not to share personal details to avoid privacy concerns [66]. Besides installing the MyWarung© app, the intervention using the in-app modules was conducted via the WhatsApp chat group, call, and automated daily notification regarding the information provided in MyWarung©. Figure 5 displays the setting and schedule of the intervention given.

To emphasise the application of each module, they were distributed separately every week in WhatsApp’s chat group in the form of gentle reminders to access the app. The 12-week intervention period was sufficient to cover all seven modules together with the submodules provided in the MyWarung© app. A minimum one-week intervention was utilised for each science-based knowledge module. Meanwhile, the food poisoning prevention modules required approximately two weeks each to obtain a total of five submodules.

Roberts et al. [67] evaluated behavioural changes in parents’ safety and actions using a mobile-app-based intervention. The Make Safe Happen© app was implemented for a 7-to-10-day intervention period, with users receiving information on preventing unintentional child harm for at least one week. After using the app during the intervention period, participants were more conscious of making their homes safer for their children. Similarly, the current study adopted a one-week intervention by disseminating the information given in MyWarung© app through the WhatsApp’s chat group. Since practical-based interventions have a significant impact on improving preventive behaviour among consumers [55], the intervention period was extended beyond the one-week science-based knowledge distribution: two weeks for selecting BeSS food premises, three weeks for food premise assessment, and two weeks to explore the channel to file complaints to the Malaysian Ministry of Health when food poisoning occurred or when witnessing a dirty food premise after dining out.

This step was taken to ensure the participants focused on each in-app module. Furthermore, a discussion between researchers and participants was carried out after every module was distributed. Consumers were asked to indicate the extent of implementation and execution of food poisoning preventive practices after referring to MyWarung©’s modules. These sessions also allowed researchers to address inquiries from participants regarding the modules via WhatsApp’s chat group and encourage the participants to implement the information given in MyWarung© and practice food poisoning prevention behaviour when eating out.

Tong et al. [68] stated that 16 reviews used a 12-week intervention period to yield significant outcomes in the intervention group relative to the control group. In addition, several intervention studies showed significant improvements in preventive behaviour after implementing a 12-week intervention duration [34,69,70,71]. As a result of the significant improvement and its suitability to the number of modules offered in the MyWarung© app, the current study adopted a 12-week educational period for respondents. Nevertheless, a longer intervention duration could be more effective because it is reasonable to assume that the longer the interventional duration, the more respondents might learn and change their habits.

### 2.7. Outcome Measures

The primary outcomes of the current study include the KAP2 of food poisoning and its prevention. In addition, both pre-and post-intervention assessments among intervention and control groups were evaluated.

### 2.8. Data Analysis

The data were analysed using Statistical Package for the Social Sciences (SPSS) software version 22. The data were presented as frequencies with percentages for nominal variables and median and interquartile range (IQR) for numerical variables. Since all scores’ normality was not met using the Kolmogorov–Smirnov test at a significance level of 5%, the data analysis was performed using a non-normal distribution with median (IQR) analysis. The Mann–Whitney test was used to measure the median score of inter-groups, while the Wilcoxon sign rank test was used to assess the effects of the median scores of KAP2 on food poisoning, before and after the intervention for both groups.

## 3. Results

### 3.1. Sociodemography

The majority of respondents in the intervention group were female (58.0%), age 21–30 years (34.0%), married (58.0%), and self-employed (30.0%). Meanwhile, most participants in the control group were female (70.0%), 41–50 years old (34.0%), married (82.0%), and unemployed (64.0%). The respondents in both groups were Malay, and 42.0% of the intervention group and 60.0% of the control group had an income of less than RM500 since they were unemployed (Table 1).

### 3.2. Between-Group Differences

The Mann–Whitney test was used to measure the intervention’s inter-group results. Table 2 demonstrated that before the intervention, there was a significant difference between the intervention and control group on the median knowledge score (*p* = 0.000), attitude (*p* = 0.008), and practice (*p* = 0.010) towards prevention of food poisoning. Meanwhile, for other subdomains of perceived barriers and susceptibility, the scores revealed no significant differences between the intervention and control groups before and after the intervention.

### 3.3. Within-Group Differences

#### 3.3.1. Effect of an Intervention Programme on Knowledge towards Food Poisoning and Its Prevention

Wilcoxon sign rank test was used to determine the within-group effects of the intervention. Based on Table 3, there was a significant increase in food poisoning knowledge (*p* = 0.000) with a pre-intervention median score of 27.5 (IQR 10.0) to a post-intervention median score of 30.0 (IQR 8.0).

#### 3.3.2. Effect of an Intervention Programme on Food Poisoning Preventive Attitude

The findings on food poisoning preventive behaviour are reported in Table 4. There was a significant increase in food poisoning preventive attitude (*p* = 0.001), with a pre-intervention median score from 43.0 (IQR 7.0) to a post-intervention median score of 46.5 (IQR 6.0).

#### 3.3.3. Effect of an Intervention Programme on Food Poisoning Preventive Practice

Table 5 shows a significant increase in food poisoning preventive practice during dining out (*p* = 0.000), with a median score of 33.0 (IQR 6.0) to 35.0 (IQR 5.0) among the intervention group.

#### 3.3.4. Effect of an Intervention Programme on Food Poisoning Preventive Perception

The intervention using MyWarung© mobile application in the current study did not significantly increase perceived barriers and susceptibility in both the intervention and control groups. However, there is a significant increase in perceived susceptibility among the control group with a median score of 8.0 (IQR 2.0) before the intervention to 10.0 (IQR 2.0) post-intervention (Table 6).

## 4. Discussion

Significant differences were found in knowledge, attitude, and practice scores between the intervention and control groups before the intervention programme. The control group’s median scores were higher than the intervention group, indicating that the control group’s level of knowledge, attitude, and practice was higher than the intervention group before the intervention programme. Possible explanations for these disparities include a demographic variation in the study population and the study location for data collection [72]. In the present research, the study population resided in different locations; thus, the data collection for intervention and control groups were drawn from separate villages.

Furthermore, food poisoning preventive knowledge and behaviour vary between individuals, contributed by their personal experience before any intervention programme is implemented. According to Osaili et al. [73], there was a significant association between Jordanian consumers who had experienced food poisoning with their level of food safety knowledge, where respondents with food poisoning experience recorded higher scores compared to those who did not. This is self-evident, so in future studies, a question on the food poisoning experience topic should be included in the questionnaire. Furthermore, people who have suffered from food poisoning tend to be more knowledgeable about food safety since they may have sought further information to protect themselves from future occurrences [74].

Moreover, respondents’ education levels may have led to the varying levels of food poisoning KAP2 in the intervention and control groups. The higher the educational level, the better the preventive knowledge and behaviour among respondents [73,75,76]. However, the current study showed that the intervention group had more respondents with higher educational levels than the control group.

Besides, gender also impacted food poisoning preventive knowledge and behaviour, resulting in the discrepancy between the intervention and control groups. In the current study, there were more female participants in the control group than the intervention group. Previous KAP studies reported that females tend to have higher knowledge, attitude, and practice in food poisoning prevention than males [77,78]. Similarly, Malhotra et al. [79] conducted a study to evaluate the impact of a health education intervention on knowledge and attitude towards food, personal hygiene, and self-reported handwashing practices and found that female respondents exhibited higher knowledge about food hygiene practices since they were routinely involved in food preparation.

Apart from that, respondents’ age group also caused a variation in knowledge, attitude, and practice scores between the intervention and control groups in the current study. The control group had more older respondents than the intervention group. Mohd Firdaus et al. [80] reported that knowledge levels increased with age. However, Shahar et al. [81] conducted a study among older Malaysian adults in rural and urban areas and discovered that older adults in rural areas had a lower level of knowledge because they commonly face higher risks of health problems due to lower accessibility to proper health care treatment than urban residents. Therefore, although previous studies have reported a high level of knowledge among the elders, an education programme for young and older consumers should not be neglected. Health education is essential at all ages, especially for the younger generation, since learning health-related knowledge, attitudes, and behaviour should begin at an early age [82].

The overall outcome in this study showed that the intervention programme has successfully educated the consumers in the intervention group on the KAP2 of food poisoning. Similarly, earlier studies have demonstrated that health-related smartphone app usage significantly improved participants’ knowledge, attitude, and practice towards disease and preventive behaviour [64,65,83]. These results, however, are contrary to Ismail et al. [60], who reported no significant increase in disease preventive behaviour among respondents after using the smartphone app as an intervention tool. This inconsistency may have resulted from the lack of self-awareness and obstacles in practising health preventive behaviour in their daily life. Nevertheless, the availability of health-promoting mobile apps has increased dramatically in recent years, thus improving preventive behaviour among users [84]. Furthermore, this technological development allows users to revisit health apps by simply clicking on them, completing security actions as often as desired, and setting reminders for future actions [32].

Likewise, mobile apps as health intervention methods have been proven by many studies to substantially improve respondents’ knowledge and preventive behaviour of diseases [66,85,86]. When users are regularly exposed to the content at their fingertips [87], the updated in-app information will improve the user knowledge, resulting in longer knowledge retention on certain diseases over time [88]. Therefore, the mobile application is an ideal platform to deliver both simple and effective interventions. The present study proved that the MyWarung© app increased food poisoning-preventive behaviour practices among consumers when dining out.

Nevertheless, the MyWarung© mobile app did not receive positive feedback as it increased perceived barriers and susceptibility to food poisoning prevention when eating out. Similarly, Ismail et al. [60] found no significant change in health preventive behaviour perception among mobile app users. Thus, more effort is needed to reduce the user’s perceived barriers and susceptibility to improve their likelihood of implementing preventive behaviours in their daily life.

Regarding the methodology used in the education programme to increase disease-preventive behaviour, information dissemination in combination with practical features was proven effective. The finding is supported by Nik Rosmawati et al. [54], who reported that using posters alone as an education tool or education without hands-on training has failed to improve handwashing practices. Therefore, besides including education by providing posters and videos in the MyWarung© app, the users were also offered practical features that focused on improving their food poisoning preventive practices. Besides, Idris et al. [88] mentioned that interventions should be based on the culture and beliefs of a population to improve and maintain positive attitudes and practices. Therefore, the contents of the MyWarung© app were developed in *Bahasa Melayu* to convey medical information among the community since most Malaysians, especially those from Terengganu, are more comfortable with the language.

## 5. Conclusions

In summary, the present study found that MyWarung© mobile application usage effectively influenced consumer knowledge, attitude, and practice on food poisoning and its prevention. However, further research in smartphone app development on the Android and iOS platforms is required to educate more consumers in the future. Furthermore, the current study was limited to Terengganu; thus, additional research should be conducted in other Malaysian states to provide more input and educate more consumers on how to avoid food poisoning when dining out.

## Figures and Tables

**Figure 1 ijerph-18-10294-f001:**
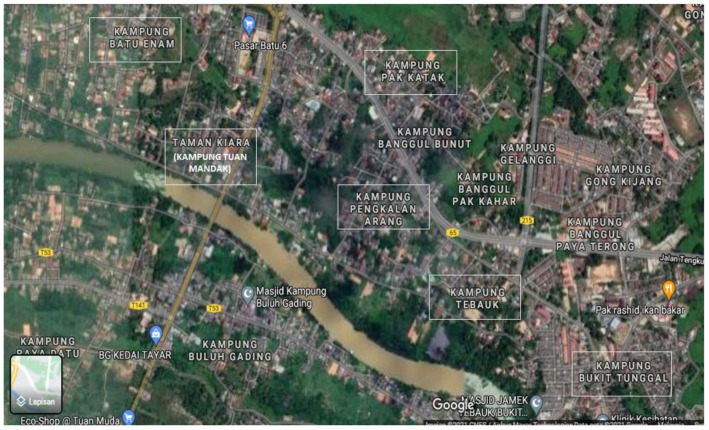
The location of six selected villages.

**Figure 2 ijerph-18-10294-f002:**
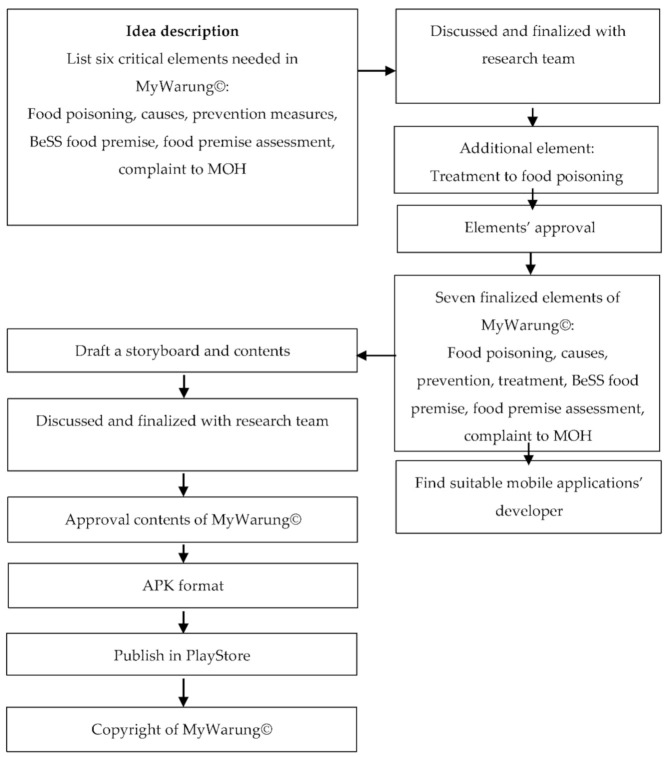
Development process of MyWarung©.

**Figure 3 ijerph-18-10294-f003:**
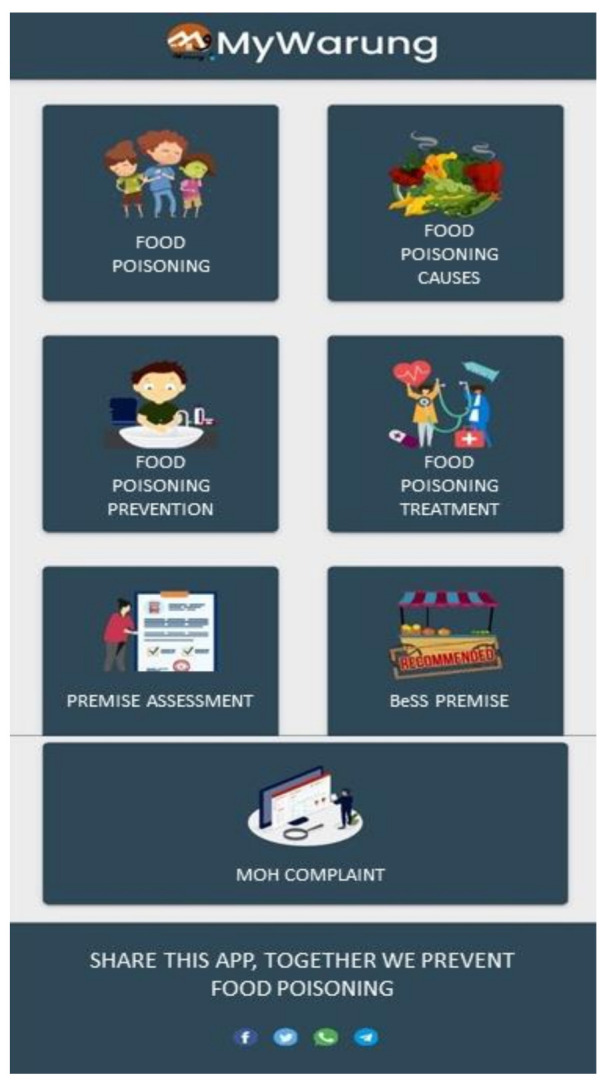
Landing page of developed MyWarung© mobile application.

**Figure 4 ijerph-18-10294-f004:**
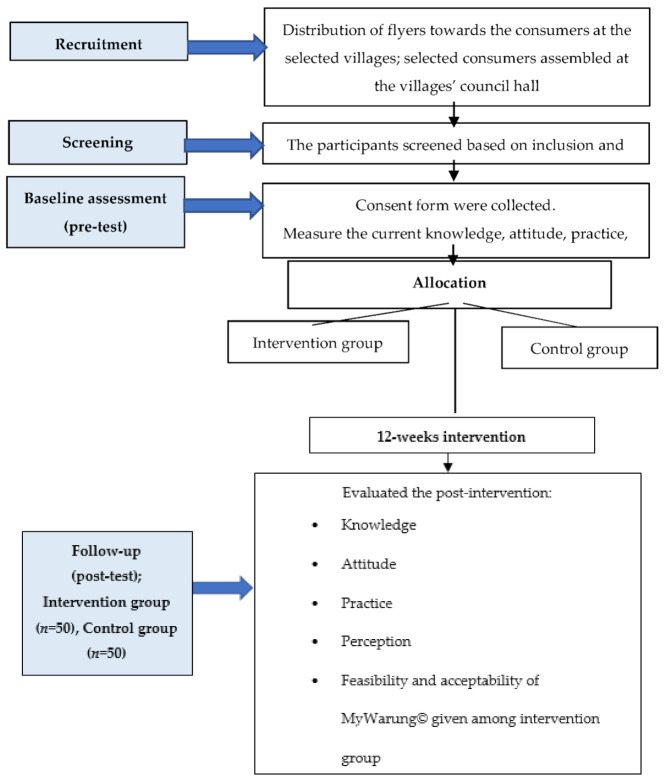
Flow diagram of the study.

**Figure 5 ijerph-18-10294-f005:**
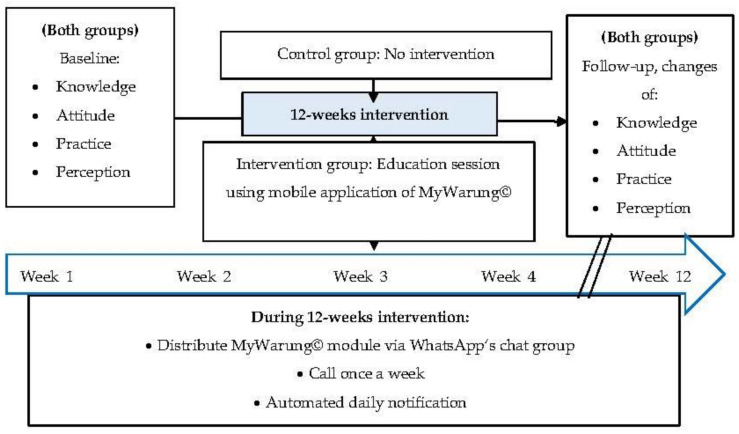
Setting and schedule of the intervention.

**Table 1 ijerph-18-10294-t001:** Socio-demographic data of the respondents (*n* = 100).

Socio-Demographic Profile	Groups			
	Intervention (*n* = 50)		Control (*n* = 50)	
	*n*	%	*n*	%
Gender				
Male	21	42.0	15	30.0
Female	29	58.0	35	70.0
Age group				
18–20 years	7	14.0	2	4.0
21–30 years	17	34.0	10	20.0
31–40 years	11	22.0	8	16.0
41–50 years	3	6.0	17	34.0
>50 years	12	24.0	13	26.0
Ethnicity				
Malay	50	100.0	50	100.0
Marital status				
Single	18	36.0	7	14.0
Married	29	58.0	41	82.0
Separated/divorced/widowed	3	6.0	2	4.0
Academic qualification				
Informal education	0	0.0	1	2.0
Primary school	4	8.0	0	0.0
Secondary school	20	40.0	28	56.0
Certificate/STPM/A level/GCE/ Foundation/matriculation/ diploma	18	36.0	12	24.0
Tertiary education (degree/master’s/ PhD)	8	16.0	9	18.0
Job sector				
Self-employed	15	30.0	7	14.0
Government sector	10	20.0	2	4.0
Private sector	11	22.0	9	18.0
Unemployed	14	28.0	32	64.0
Income level				
RM0–RM500	21	42.0	30	60.0
RM501–RM1000	11	22.0	6	12.0
RM1001–RM1500	10	20.0	5	10.0
RM1501–RM2000	3	6.0	2	4.0
>RM2000	5	10.0	7	14.0

**Table 2 ijerph-18-10294-t002:** The score differences in the knowledge, attitude, practice, and perceived barrier susceptibility before and after intervention between groups (*n* = 100).

Variables	Intervention Group (*n* = 50)	Control Group (*n* = 50)	Z Statistics	*p*-Value
	Median Score (IQR)	Median Score (IQR)		
Knowledge				
Pre	27.5 (10.0)	30.0 (7.0)	−3.565	0.000 *
Post	30.0 (8.0)	31.0 (6.0)	−1.274	0.203
Attitude				
Pre	43.0 (7.0)	46.0 (5.0)	−2.671	0.008 *
Post	46.5 (6.0)	46.0 (6.0)	−0.410	0.682
Practice				
Pre	33.0 (6.0)	34.0 (3.0)	−2.571	0.010 *
Post	35.0 (5.0)	36.0 (8.0)	−0.964	0.335
Perceived barrier				
Pre	9.0 (3.0)	9.5 (3.0)	−0.805	0.421
Post	10.0 (6.0)	10.0 (4.0)	−0.076	0.939
Perceived susceptibility				
Pre	8.0 (2.0)	8.0 (2.0)	−0.056	0.955
Post	10.0 (2.0)	10.0 (2.0)	−0.294	0.768

* Mann–Whitney test; significance level at *p* < 0.05. Knowledge score (minimum = 0 points, maximum = 42 points). Attitude score (minimum = 10 points, maximum = 50 points). Practice score (minimum = 10 points, maximum = 40 points). Perceived barrier score (minimum = 3 points, maximum = 15 points). Perceived susceptibility score (minimum = 2 points, maximum = 10 points).

**Table 3 ijerph-18-10294-t003:** Knowledge score before and after intervention among respondents (*n* = 100).

Variables	Intervention Group (*n* = 50)			Control Group (*n* = 50)		
	Pre	Post	*p*-Value	Pre	Post	*p*-Value
	Median Score (IQR)	Median Score (IQR)		Median Score (IQR)	Median Score (IQR)	
Knowledge	27.5 (10.0)	30.0 (8.0)	0.000 *	30.0 (7.0)	31.0 (6.0)	0.324

* Wilcoxon sign rank test; significance level at *p* < 0.05. Knowledge score (minimum = 0 points, maximum = 42 points).

**Table 4 ijerph-18-10294-t004:** Food poisoning preventive attitude score before and after intervention among respondents (*n* = 100).

Variables	Intervention Group (*n* = 50)			Control Group (*n* = 50)		
	Pre	Post	*p*-Value	Pre	Post	*p*-Value
	Median Score (IQR)	Median Score (IQR)		Median Score (IQR)	Median Score (IQR)	
Attitude	43.0 (7.0)	46.5 (6.0)	0.001 *	46.0 (5.0)	46.0 (6.0)	0.490

* Wilcoxon sign rank test; significance level at *p* < 0.05. Attitude score (minimum = 10 points, maximum = 50 points).

**Table 5 ijerph-18-10294-t005:** Food poisoning preventive practice score before and after intervention among respondents (*n* = 100).

Variables	Intervention Group (*n* = 50)			Control Group (*n* = 50)		
	Pre	Post	*p*-Value	Pre	Post	*p*-Value
	Median Score (IQR)	Median Score (IQR)		Median Score (IQR)	Median Score (IQR)	
Practice	33.0 (6.0)	35.0 (5.0)	0.000 *	34.0 (3.0)	36.0 (8.0)	0.131

* Wilcoxon sign rank test; significance level at *p* < 0.05. Practice score (minimum = 10 points, maximum = 40 points).

**Table 6 ijerph-18-10294-t006:** Perceived barrier and perceived susceptibility score before and after intervention among respondents towards food poisoning preventive behaviour during dining out (*n* = 100).

Variables	Intervention Group (*n* = 50)			Control Group (*n* = 50)		
	Pre	Post	*p*-Value	Pre	Post	*p*-Value
	Median Score (IQR)	Median Score (IQR)		Median Score (IQR)	Median Score (IQR)	
Perceived barrier	9.0 (3.0)	10.0 (6.0)	0.129	9.50 (3.0)	10.0 (4.0)	0.422
Perceived susceptibility	8.0 (2.0)	10.0 (2.0)	0.069	8.0 (2.0)	10.0 (2.0)	0.012 *

* Wilcoxon sign rank test; significance level at *p* < 0.05. Perceived barrier score (minimum = 3 points, maximum = 15 points). Perceived susceptibility score (minimum = 2 points, maximum = 10 points).

## Data Availability

The data presented in this study are available within the article.

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
