# Peer review of "Effect of Smartphone App’s Intervention on Consumers’ Knowledge, Attitude, Practice, and Perception of Food Poisoning Prevention When Dining Out at Selected Rural Areas in Terengganu"

_ijerph, 2021, doi:10.3390/ijerph181910294_

Round 1
Reviewer 1 Report
The food safety and the prevention of food borne outbreaks are a crucial issue for worldwide. In this paper, the Authors proposed an interesting study on effect of a Smart Phone - Apps Intervention on Consumers’ knowledge, attitude, practice, and perception of food poisoning prevention during dining out at selected rural area in Terengganu.
The introduction described exhaustively the aim of research, the research methodology was detailed, but there were some points which I suggest revising:
- Line 138-143 results too repetitive
- Add a bracket line 159….. (34)
- Par. 2.6 - 2.7 seems repetitive.
Results were stated clearly.
Discussion and conclusion were well argued.
Author Response
Thank you very much for your comments. All has been changes as suggested.
- Add a bracket line 159: A quasi-experimental pre-and post-intervention study was conducted because this was a population-based study in which it was not feasible to randomize at the individual level (34).
- Subtopic 2.5 Recruitment and 2.6 Data collection has been combined, and repeated sentences have been deleted in the revised manuscript.

Reviewer 2 Report
The study gives good insight on the possibility that educational campaign for food safety can be approached trough simple app in smart phone. However, I have some great concern on the study design targeting "rural" community for this issue. Why did you pose so much attention in rural and not in general population? You stated that the region was chosen because a larger number of food poisoning cases occurred in 2018 but were these cases more frequent in rural population? You also stated that the region is partially rural since towns with industrial and administrative offices areas are also present. Some wrong practices in terms of food safety are very common among general population and not limited to rural especially if you are referred to the habit of eating out. After all, I think that your app can be useful to everybody going out for lunch or dinner.
Specific comments:
Introduction :Food poisoning refers to diseases caused by infectious organism foods consumption but also to the toxins produced by the infectious organisms.
Previous study by Abdullah & Ismail (23) had conducted a study??? line 75
STUDY location section.
"The overall district of Kuala Nerus can be indicated as not a rural area because it consists of business facilities, residential and leisure activities with a modern and attractive urban environment" (Line129). So at this point you should justify why did you chose specifically the rural areas of this district.
From lane 132 "Among the developed infrastructures in Kuala Nerus was GongBadak Industrial Area; Institute of Higher Learning including Universiti Malaysia Terengganu (UMT), Universiti Sultan Zainal Abidin (UNISZA), Industrial Training Institute (ILP), Institute of Teacher Education Dato Razali Ismail Campus and others; Airport; Sports Complex; Military Camp; Tabung Haji Terminal Complex; and UNISZA Teaching Hospital is under construction. In general, the development at the Kuala Nerus District Administrative Center is the "Modern & Green City" concept (37). I do not think that it's necessary to describe in such details the region in aspects that are not important for the research and actually are in contrast with rural definition.
lane 149 Therefore, the villages in Kuala Nerus with a population of less than 10,000 as a rural area were chosen as a sampling area in the present study. This is not a "scientific" definition for rural and you explain much better in the following section the characteristics of the villages included in the study so I think that you can delete this sentence from here.
Study sampling
Figure 1: Sampling frame of the study . This figure is actually a diagram of teh administartive organization of the study location starting from the Kuala Terengganu city Council ? But for the aim of the study I think is not relevant while it could be important to have a map with the study area indicatthe villages that you choose as rural areas to select the groups.
What are the inclusion-exclusion criteria? I think is not clear in the manuscript .
In figure 3 could be nice to add some English translation on the different figures of the app just for international readers.
Also you should discuss why did you make a 12 weeks period for the intervention. Considering that it is possible that many respondents eat outside only once a week, probably a longer period could be better because it is reasonable to assume that the more you use the app the more you learn and change your habit.
Discussion: "... with those who reported a previous experience of food poisoning having a higher score of knowledge compared to those who did not". This is absolutely obvious so a question related to this topic should be included in the questionnaire in future studies.
Idris et al. (83) is repeated twice in the reference list
Author Response
Thank you very much for your comments.
- Study Location = Rural areas in Terengganu has been selected as study location in the current study due to few factors and supported by previous studies which have been described in manuscript as following: (lane 76-88) Previous study by Abdullah & Ismail (23) had conducted a cross-sectional study on the food poisoning outbreak in Terengganu and mentioned that rural areas in Terengganu are at a higher risk of food poisoning outbreaks, showing food poisoning prevention in rural areas, particularly in Terengganu is vital. Besides, lack of food poisoning knowledge, preventive attitudes and practices were found among the rural communities as reported by Bisholo et al. (8) which food that was sold at many informal food outlets and street vendors in rural areas causing public health concern, as only 17.2% of rural consumers concern on food safety of food prepared away from home, while 52.9% of them were not concerned at all. Poor food safety practices among rural consumers also were reported by Adebowale & Kassim (24). These previous studies showed that the problem on food poisoning cases among rural communities become more severe due to lack of public awareness on food poisoning and its prevention, indicating the educational program is needed among consumers in the rural area. Therefore, it is important to conduct an educational approach among rural consumers particularly focusing on food poisoning prevention during dining out. (lane 577-580) Shahar et al. (82) conducted a study among Malaysian older adults in rural and urban areas and reported that rural older adults had a low level of knowledge because they commonly face a higher risks of health problems due to less accessibility to proper health care treatment compared to urban residents.
- The suggestion sentence has been written in revised manuscript as following (lane 36-37): Food poisoning refers to disease caused by infectious organism foods consumption but also to the toxins produced by the infectious organisms (1).
- The correction has been written in the manuscript as following (lane 77): The previous study by Abdullah & Ismail (23) had conducted a cross-sectional study on the food poisoning outbreak in Terengganu.
- Lane 132 "Among the developed infrastructure in Kuala Nerus was Gong Badak Industrial Area; Institute of Higher Learning including Universiti Malaysia Terengganu........."Modern & Green City" concept (37)" - This sentence had been deleted as suggested.
- Lane 149 Therefore, the villages in Kuala Nerus with a population of less than 10,000 as a rural area were chosen as sampling areas in the present study - this sentence has been deleted as suggested.
- The sampling frame of study has been deleted as suggested. The maps with study areas indicate the chosen villages as study location has been added in the revised manuscript, as suggested.
- The inclusion-exclusion criteria of participants have been added in the revised manuscript as following (lane 224-228): The participants were inclusive of foods purchased outside at least once a week; ages 18 and over; readiness for 12 weeks of study; willingness to used mobile app given if selected in intervention study; and the ability to read and write in Malay as a questionnaire provided in the Malay language. Furthermore, the exclusion criteria for selecting the participants were refusal to participate and inability to consent from participants.
- English translation of the lading page of the developed mobile application has been added.
- Discussion on choosing 12-weeks intervention duration has been added as following: A 12-week intervention period was implemented in the current study. The study by Tong et al. (68) stated that 16 reviews used a 12-weeks intervention period, indicating the average duration of educational intervention studies was 12 weeks and showed a significant change in the intervention group after 12 weeks. In addition, a numbers of intervention studies showed a significant improvement in preventive behavior after implementing a 12-weeks intervention duration (69,70,71,72). As a results of the positive significant improvement and its suitability on the number of modules offered in MyWarung mobile application, the current study adopted a 12-weeks educational period among respondents. Also, a longer interventional duration could be better because it is reasonable to assume that the longer interventional duration, the more respondents can learn and change their habits.
- "This is absolutely obvious so a question related to this topic should be included in the questionnaire in future studies"- This sentence has been added in the revised manuscript as suggested (lane 572-573).
- The repeated reference has been deleted. Additional references have been added to the revised manuscript.

Round 2
Reviewer 2 Report
line 45: Most food poisoning bacteria are Salmonella spp., Escherichia coli, Campylobacter, Staphylococcus aureus, and Clostridium botulinum (bacteria in italics )
In the introduction food poisoning is repeated several times , For example
A lack of awareness of the severity of this disease continues to increase food poisoning in Malaysia (5). The highest incidence rate recorded for food poisoning cases was 45.71 per 100,000 Malaysians in 2018 (6). The increase in food poisoning cases reported over the years reflects the poor food safety situation in Malaysia and increases the burden of foodborne disease (7).
It could be changed in : "A lack of awareness of the severity of this disease continues to increase food poisoning cases in Malaysia (5) with the highest incidence rate of 45.71 per 100,000 Malaysians recorded in 2018 (6). The increase in food poisoning cases reported over the years reflects the poor food safety situation in Malaysia (7)."
Author Response
The manuscript has been submitted for proofreading. All have been changes as suggested as following:
- Salmonella spp., Escherichia coli, Campylobacter, Staphylococcus aureus, and Clostridium botulinum (bacteria in italics)
- A lack of awareness of the severity of this disease continues to increase food poisoning cases in Malaysia (5) with the highest incidence rate of 45.71 per 100,000 Malaysians recorded in 2018 (6). The increase in food poisoning cases reported over the years reflects the poor food safety situation in Malaysia (7)."
